# QPO: Query-dependent Prompt Optimization via Multi-Loop Offline Reinforcement Learning

**Yilun Kong[1]***, **Hangyu Mao[2]†, Qi Zhao[1], Bin Zhang[34], Jingqing Ruan[34], Li Shen[5],
Yongzhe Chang[1], Xueqian Wang[1]†, Rui Zhao[2], Dacheng Tao[6]**

[1] *Tsinghua University;* [2] *SenseTime Research;* [3] *Institute of automation,Chinese academy of science;*
[4] *School of Artificial Intelligence,University of Chinese Academy of Sciences;* [5] *Sun Yat-Sen University;*
[6] *Nanyang Technological University*

**Reviewed on OpenReview:** *https://openreview.net/forum?id=bqMJToTkvT*

## Abstract

Prompt engineering has demonstrated remarkable success in enhancing the performance of large language models (LLMs) across diverse tasks. However, most existing prompt optimization methods only focus on the task-level performance, overlooking the importance of query-preferred prompts, which leads to suboptimal performances. Additionally, these methods rely heavily on frequent interactions with LLMs to obtain feedback for guiding the optimization process, incurring substantial redundant interaction costs. In this paper, we introduce **Q**uery-dependent **P**rompt **O**ptimization (**QPO**), which leverages multi-loop offline reinforcement learning to iteratively fine-tune a small pretrained language model to generate optimal prompts tailored to the input queries, thus significantly improving the prompting effect on the large target LLM. We derive insights from offline prompting demonstration data, which already exists in large quantities as a by-product of benchmarking diverse prompts on open-sourced tasks, thereby circumventing the expenses of online interactions. Furthermore, we continuously augment the offline dataset with the generated prompts in each loop, as the prompts from the fine-tuned model are supposed to outperform the source prompts in the original dataset. These iterative loops bootstrap the model towards generating optimal prompts. Experiments on various LLM scales and diverse NLP and math tasks demonstrate the efficacy and cost-efficiency of our method in both zero-shot and few-shot scenarios.

## 1 Introduction

Large Language Models (LLMs) have exhibited impressive prowess in various domains of natural language processing (NLP) (Ouyang et al., 2022; Touvron et al., 2023; Achiam et al., 2023). Prompt engineering, a method that simply adds an instruction to the input query, emerges as a lightweight and promising solution for adapting LLMs to downstream tasks without the need for parameter tuning (Liu et al., 2023a; Ajith et al., 2023). Since the performance of LLMs towards a particular task is significantly influenced by the quality of the prompt, the key challenge of prompting lies in how to design the optimal prompts.

Numerous prompt engineering algorithms have been proposed in recent years. Some algorithms (Zhou et al., 2022; Wang et al., 2023b; Guo et al., 2023; Wang et al., 2024b) leverage LLMs as a prompt optimizer, employing black-box optimization to derive the best prompts. Others utilize reinforcement learning (RL) (Deng et al., 2022; Zhang et al., 2022; Kong et al., 2024) to train a policy model to generate the optimal prompts. Despite these advances, prompt engineering still grapples with great challenges:

---

*Work done during an internship at SenseTime Research.
†corresponding authors: Hangyu Mao<maohangyu@sensetime.com>; Xueqian Wang<wang.xq@sz.tsinghua.edu.cn>

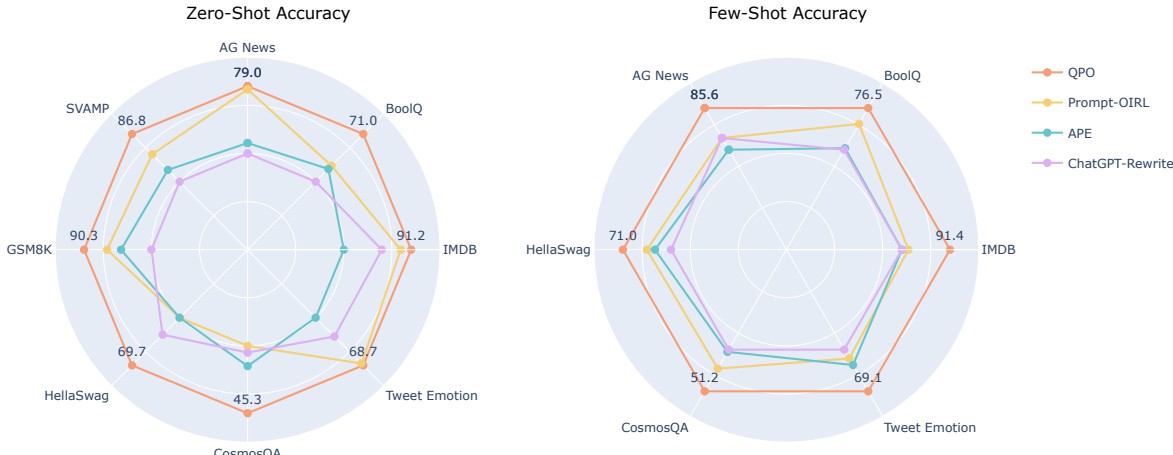

Figure 1: QPO delivers state-of-the-art performance across a wide variety of tasks, superior to existing query-dependent and query-agnostic prompt optimization methods.

(1) Most of previous algorithms only focus on obtaining task-level optimal prompts, aiming to attain an optimal average performance across all queries within a specific task. However, they overlook the critical insight that no single prompt can be ideal for every possible query (Sun et al., 2023).

(2) All the aforementioned methods require frequent interactions with the target LLMs to obtain feedback on the current prompts to guide optimization, incurring significant costs due to the high expense of inferences with target LLMs.

In fact, query-level optimal prompts can yield better performance (Zhang et al., 2022). Meanwhile, offline optimization can serve as a promising method for reducing the costs associated with LLM interactions based on the existence of a wealth of prompt optimization datasets, which can be easily available as by-products during the evaluation of existing prompting strategies. However, research on the two important domains remains critically scarce.

To tackle the above challenges, we propose **QPO**, a **Q**uery-dependent **P**rompt **O**ptimization method through multi-loop offline reinforcement learning. The overall process of our framework is illustrated in Figure 2. Firstly, we employ offline RL for the initial loop to fine-tune a small-scale pretrained language model (PLM) as a policy model on the pre-collected dataset, enabling it to generate specific prompts based on input queries and given rewards for the target LLM. Subsequently, we utilize the trained policy model to efficiently explore new queries and prompts to augment the dataset, as the generated novel prompts are supposed to be more suitable for the current queries than other prompts in the dataset. This process eliminates the necessity to evaluate all collected prompts for each query, while suffices to assess only a few query-specific high-quality prompts, significantly reducing the required interactions with the target LLM. The augmented dataset, enriched with more queries and superior prompts, can further train the policy model and enhance its performance. Consequently, we establish the iterative process wherein the policy model, fine-tuned via offline reinforcement learning, efficiently generates improved data, which is then utilized to further fine-tune the model, creating a cyclical feedback loop. This bootstrapping learning process can achieve substantial improvements in limited loops with minimal interactions with the LLMs.

We evaluate our method across different LLM scales on various neural language understanding and math reasoning tasks using both zero-shot and few-shot settings. As shown in Figure 1, QPO reaches state-of-the-art performance, superior to both query-dependent and query-agnostic prompting methods. Compared to online optimization approaches, our method only needs a minimal number of interactions with the target LLM to augment the dataset after each loop of offline optimization, which strikes a great balance between performance and LLM interaction costs. In addition, we perform extensive ablations on different aspects

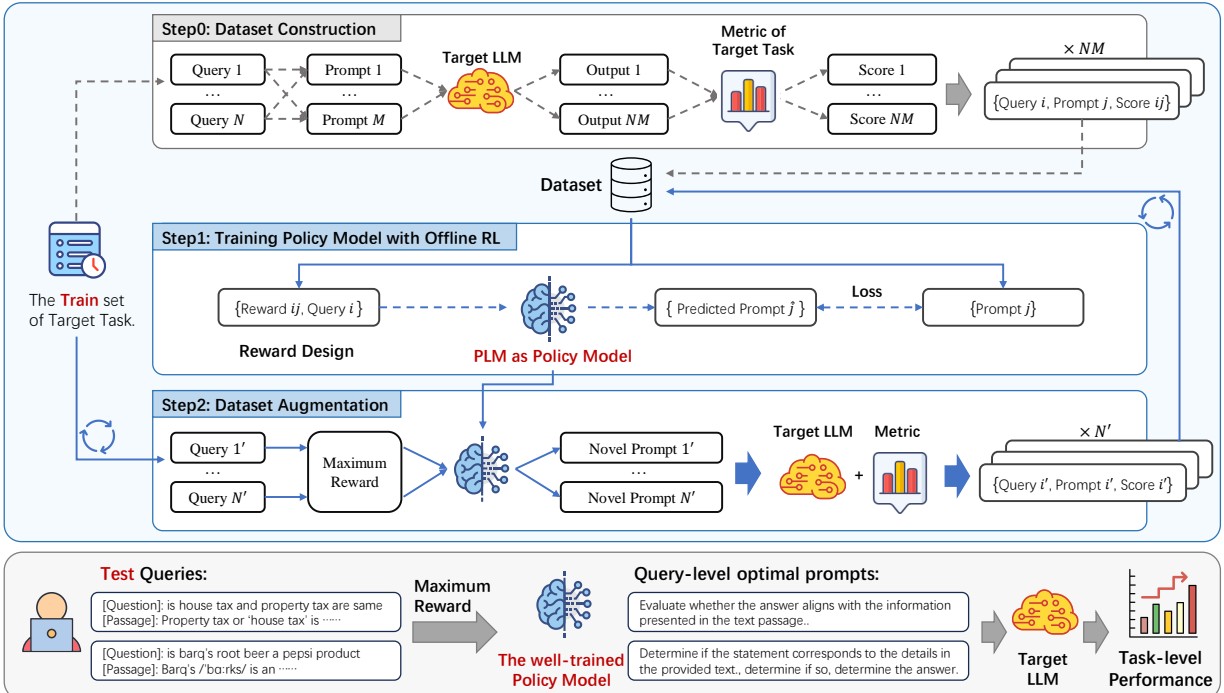

Figure 2: The overview of our framework. We aim at training a small pretrained language model (PLM) as the policy model, which can generate query-level optimal prompts based on the given queries, and ultimately improve the target LLM's task-level performance. In the training phase, depicted by the blue frame above, the blue arrows represent our primary iterative process, while the gray arrows indicate the initial steps. Specifically, in Step0, the offline dataset is constructed as a by-product of evaluating existing prompts. The below frame with gray background indicates the testing phase.

of the proposed algorithm. The policy model demonstrates great cross-model generalization ability, further enhancing the value of our approach. In conclusion, our contributions are three-fold:

- We identify the overlooked query-dependent prompt optimization problem, and introduce an Offline RL paradigm to fine-tune a PLM to generate query-specific optimal prompts.

- We propose a comprehensive framework, QPO, which leverages the existence of offline datasets and employs a novel Multi-Loop Augmentation technique to facilitate data augmentation with minimal interactions, and ultimately bootstrap the performance of prompt optimization through offline RL fine-tuning.

- We benchmark our method across various LLMs on extensive datasets, demonstrating its superiority and great potential in broader applications.

## 2 Methodology

To automatically generate query-level optimal prompts in a cost-efficiency method, we propose QPO, a novel query-dependent prompt optimization framework. The overall process is shown in Figure 2. In the first round, QPO harnesses the readily existing datasets to fine-tune a pre-trained language model into a policy model through offline reinforcement learning, enabling it to generate query-specific optimal prompts based on input queries. Subsequently, this policy model efficiently explores a broader query space and enriches the prompt space with more diverse and high-quality prompts, both of which contribute to the augmentation of the dataset. The augmented dataset, in turn, further refines the policy model for the next loop. Consequently, through multi-loop training and data augmentation, the performance of prompt optimization is bootstrapped for improvement.

### 2.1 Problem Formulation

This study focuses on training the policy model to generate prompts, which can instruct a target LLM to output expected answers that fulfill the query-dependent objective, based on the given queries.

**Query and Answer.** We consider the task of answering queries $q \in \mathcal{Q} = \mathcal{V}^\infty$ expressed in a natural language, where $\mathcal{V}$ denotes the vocabulary. Each query $q$ is supposed to have an expected answer $y^* \in \mathcal{Y}$ as ground truth. Queries annotated with ground-truth answers can be employed as demonstrations $d$ for few-shot scenario.

**Prompt and Policy Model.** The performance of an LLM can be significantly enhanced through appropriate prompts. The prompt $p \in \mathcal{P} = \mathcal{V}^\infty$ is a natural language instruction that explains to the LLM how to complete the query and output the predicted answer. In this paper, we utilize a fine-tuned policy model $\pi : \mathcal{Q} \to \mathcal{P}$ to generate query-specific prompts based on the given queries: $\hat{p} = \pi(q)$.

**Target LLM.** These tasks are performed by employing a target LLM $\ell : \mathcal{Q} \to \mathcal{Y}$, feeding queries $q$ into the LLM to get answers. With prompts, the answers can be obtained by $\hat{y}^{(i,j,k)} = \ell(p^{(j)}, d^{(k)}, q^{(i)})$, where $q^{(i)}, p^{(j)}, d^{(k)}$ denote the $i$-th query, $j$-th prompt, and $k$-th combination of demonstrations, respectively. For zero-shot setting, $d$ is null, so we use $\hat{y}^{(i,j)}$ as a shorthand unless otherwise specified.

**Query-dependent Objective.** Given a task with queries and ground-truth answers, the quality of the predicted answers can be evaluated by a metric $\rho(y^*, \hat{y})$, such as correctness. The objective of query-dependent prompt optimization can be formulated as below:

$$\pi^* = \arg\max_\pi \rho(y^{*(i)}, \ell(\pi(q^{(i)}), q^{(i)}))_{i \in [N]}, \tag{1}$$

which aims to optimize the policy model to generate a query-specific prompt that enhance the target LLM's performance on this particular query. As its performance improves across each single query, the overall task performance will consequently improve.

### 2.2 Model Query-dependent Prompt Optimization as Offline RL

**Reinforcement Learning Formulation.** As a text generation problem, prompt generation can be formulated as a Markov Decision Process (MDP) $\langle \mathcal{S}, \mathcal{A}, r, f \rangle$ with a finite state space $\mathcal{S}$, action space $\mathcal{A}$, reward function $r$, and state-transition probability function $f$, where a sparse reward can be obtained only after the complete prompt is generated. *Instead of dedicatedly designing intricate proxy rewards, we formulate the MDP as a single-step decision-making process.* In offline RL (Levine et al., 2020), given the dataset:

$$\mathcal{D} = \{q^{(i)}, p^{(j)}, r^{(i,j)} = \mathcal{R}(q^{(i)}, p^{(j)})\}_{i \in [N], j \in [M]},$$

where the initial state $q \in \mathcal{S}$ is the input query with $n$ tokens $q = (q_1, ..., q_n)$ and the corresponding action $p \in \mathcal{A}$ represents the generated prompt with $m$ tokens $p = (p_1, ..., p_m)$, where each token $q, p$ is from the vocabulary $\mathcal{V}$. In a single-step decision episode, the policy model generates a complete prompt based on the given expected reward and query $p \sim \pi(p|r, q)$. The reward guides the quality of the generated prompt. To achieve this, we adopt Decision Transformer (DT) (Chen et al., 2021) approach to fine-tune the policy model, which is aligned with the autoregressive predicting paradigm of language models. For further discussion on why prompt generation is formulated as a single-step decision process, please refer to Appendix A.2.

**Reward Design.** In our single-step prompt generation process, the reward plays a crucial role in aligning with the true capabilities of the prompt. In this paper, we mainly focus on neural language understanding tasks and math reasoning tasks rather than generative tasks, which are challenging to assess objectively because of the poor correlation between the evaluation metrics and human preferences (Liang et al., 2022; Goyal et al., 2022). We design the reward based on two dimensions: query-level and task-level. *Query-level reward measures whether the prompt can instruct LLM to answer the specific question correctly, while task-level reward measures the prompt's average prompting ability for all queries in one task.* In particular, we adopt the average accuracy of the prompt for all test questions as the task-level reward for both zero-shot and few-shot settings,

$$\mathcal{R}_{task}(q^{(i)}, p^{(j)}) = \frac{1}{N}\sum_{i=1}^{N} \mathbb{1}\{\hat{y}^{(i,j)} = y^{*(i)}\}. \tag{2}$$

For query-level rewards, the design differs between zero-shot and few-shot scenarios due to the variations in evaluating approaches. For zero-shot evaluation, we utilize the prediction correctness as a coarse-grained measure for the prompt-query pair and additionally introduce the output perplexity (PPL) as a fine-grained penalty, which can be easily calculated by the Log-likelihood from the black-box API or directly obtained through the open-source LLM. During few-shot evaluation, the result of a query is obtained by voting or averaging across $K$ different in-context demonstration combinations to eliminate the inherent instability and randomness of few-shot setup (Gao et al., 2020; Ajith et al., 2023). We use the average correctness across demonstration combinations of the prompt-query pair as the fine-grained query-level reward in few-shot setting, which is more stable than PPL. As the query-level rewards are supposed to accurately reflect the distinctions in prompting effectiveness for various prompts across different queries, both the perplexity penalty in zero-shot setting and the averaged correctness in few-shot setting, which are decimals rather than binary integers, exhibits greater variability across different query-prompt pairs compared to the binary correctness, thus enable a more fine-grained alignment:

$$\mathcal{R}_{query}(\boldsymbol{q}^{(i)}, \boldsymbol{p}^{(j)}) = \begin{cases} \mathbb{1}\{\hat{y}^{(i,j)} = y^{*(i)}\} - \dfrac{1}{10}\text{PPL}(\hat{y}^{(i,j)}), & \text{Zero-Shot}, \\ \dfrac{1}{K}\displaystyle\sum_{k=1}^{K} \mathbb{1}\{\hat{y}^{(i,j,k)} = y^{*(i)}\}, & \text{Few-Shot}. \end{cases} \tag{3}$$

The overall reward is the sum of the query-level and task-level reward. Due to significant differences in reward scales across different tasks, we normalize the overall reward using the Min-Max Normalization method (Patro, 2015) for convenience, mapping it to the range of 0-100.

$$\mathcal{R}(\boldsymbol{q}^{(i)}, \boldsymbol{p}^{(j)}) = \mathcal{R}_{query}(\boldsymbol{q}^{(i)}, \boldsymbol{p}^{(j)}) + \mathcal{R}_{task}(\boldsymbol{q}^{(i)}, \boldsymbol{p}^{(j)}). \tag{4}$$

**Training Objective.** To enable the model to output appropriate prompts based on expected rewards and input questions, we maximize the log-likelihood with teacher forcing through DT training paradigm:

$$\mathcal{L}_{prompt} = -\mathbb{E}_{(\boldsymbol{q}, \boldsymbol{p}, r) \sim \mathcal{D}} \log p(\boldsymbol{p}|r, \boldsymbol{q}), \tag{5}$$

where we input reward as return-to-go and the complete query as state, and allow the policy model to autoregressively predict the entire prompt's tokens as an action.

To further enhance the model to distinguish the prompting abilities of different prompts for the various queries, we introduce the reward prediction loss. We directly predict the expected reward based on the output of the transformer at the first token, regarding the model as an autoencoder (Liou et al., 2014) by encoding an reconstructing the input reward. This approach can extract more information from the given reward, impose constraints on the model weights, and significantly reduce training difficulty compared to predict reward based on the output prompt at the last token. Further discussion on implementing autoencoder functionality is illustrated in Appendix A.4.

$$\mathcal{L}_r = \mathbb{E}_{(\boldsymbol{q}, \boldsymbol{p}, r) \sim \mathcal{D}} (\hat{r} - r^*)^2 \tag{6}$$

We define the overall objective function by combining the above objectives with a balancing hyperparameter:

$$\mathcal{L} = \mathcal{L}_{prompt} + \lambda \mathcal{L}_r \tag{7}$$

**Model Architecture.** To achieve the aforementioned functionality, enabling the model to fully leverage reward information and temporal dynamics in reinforcement learning, we also refine the architecture of the pre-trained language model, elevating it to serve as our policy model. We introduce a distinct reward embedding layer, separate from the natural language token embeddings, to facilitate a deeper comprehension of reward signals by the model. Furthermore, during training, we incorporate an additional reward prediction layer to implement the reward prediction loss, thereby imposing constraints on the optimization of the model's primary network. The combined enhancements in both algorithm and model structure culminate in the successful implementation of offline prompt optimization.

### 2.3 QPO Framework for Multi-Loop Augmentation

#### 2.3.1 Initial Demonstration Construction

**Dataset Construction.** We start by emphasizing the presence and significance of prompt demonstrations generated by previous research of prompt optimization. In the domain of prompt engineering, abundant prompts have been proposed to improve the LLMs' performance on downstream tasks. Generally, their efficacy is assessed at the task level. However, different queries exhibit varied preferences for different prompts. Query-level score, obtained before calculating task-level score but ignored by previous research, can play a great role in teaching the query-dependent prompting ability. As the prompts have been evaluated on open-sourced datasets, the following query-level demonstrations can be constructed as by-products. The process for constructing the offline dataset is as follows: first, we sample queries from the task dataset, then combine them with the collected prompts and input the query-prompt pairs into the target LLM to obtain answers. Through the designed reward function, we calculate the reward value for each query-prompt pair. These query-prompt-reward triplets form the samples in our offline dataset:

$$\mathcal{D} = \{\boldsymbol{q}^{(i)}, \boldsymbol{p}^{(j)}, y^{*(i)}, \hat{y}^{(i,j)}, \mathcal{R}(y^{*(i)}, \hat{y}^{(i,j)})\}_{i \in [N], j \in [M]},$$

where $M, N$ denote the number of collected prompts and queries, respectively. To enrich the initial exploration of the prompt space and simplify prompt acquisition, we leverage ChatGPT-3.5 to rewrite prompts, resulting in a diverse vocabulary and yielding a substantial number of prompts in our dataset. Details of rewriting prompts are demonstrated in Appendix B.2. While as each query requires testing with all prompts, we only utilize a small collection of queries as the initial dataset to strike a balance between algorithm performance and computational cost.

**Data Filtering.** Similar to popular offline reinforcement learning's need to differentiate expert, medium, and random datasets, in this work, we simply remove the less-quality examples with rewards below the expert threshold, defined as the 66.7th percentile of the reward range in the dataset. Notably, this approach does not discard two-thirds of the data since we use rewards as the filtering metric rather than data volume, and most samples fall within the expert interval. Almost all prompts are retained without wastage; even if a prompt performs poorly on average, it may still enable the LLM to answer correctly on specific questions, and it is retained at this query. The specific data wastage resulting from data filtering is detailed in Section 3.3.

#### 2.3.2 Multi-Loop Augmentation

After fine-tuning with the initial dataset, the policy model is supposed to generate prompts that are better suited for specific problems than any existing prompts in the dataset. Therefore, a natural and efficient approach is to use these prompts to enrich the dataset as a form of data augmentation. Moreover, as the model generates high-quality prompts tailored to specific queries, there is little concern about filtering out evaluation samples and wasting data, which presents a great opportunity to extensively assess various queries, expanding exploration of the query space.

**Query Augmentation.** As our initial dataset covers only a limited number of queries, we randomly collect both overlapped and uncovered queries from the task's training set, simultaneously focusing on optimizing prompts on existing problems and exploring unknown ones. Since only one specifically generated prompt needs to be evaluated for each problem, rather than testing all prompts, it significantly reduces computational overhead caused by the number of queries and candidate prompts. Hence, it enables large-scale exploration of new problems, avoiding the test performance degradation caused by the distribution shift of the limited initial training queries.

**Prompt Augmentation.** Based on the newly collected queries, we utilize the maximum expected reward to instruct the fine-tuned model to generate novel and superior prompts. In order to guarantee the quality and diversity of prompt exploration, we adopt the sampling generation approach with the policy model in this phase.

**Dataset Augmentation.** For augmented query-prompt pairs, the calculation of the query-level rewards remains consistent with the original formula (3), while task-level rewards are determined by the model's

overall generation capability in the current loop, that is, the average accuracy of all prompts generated in the current augmentation stage. Then, we obtain the new training examples:

$$\mathcal{D}_A = \{\boldsymbol{q}^{(i')}, \boldsymbol{p}^{(i')}, r^{(i')}\}_{i \in [N']},$$

where $N'$ denotes the number of new queries. This dataset is finally supplemented into the original dataset for the next loop's training.

After data augmentation, the updated dataset is enriched with a greater variety of queries and higher-quality prompts, which can be utilized to fine-tune the model to further improve its performance. We structure these processes into a continuous bootstrapping loop, where both offline RL and data augmentation cyclically feedback to each other. The algorithm stops when the number of iterations reaches a predefined value. Our method achieves incremental improvements utilizing offline optimization with significantly minimal interactions with LLMs. The detailed algorithm of QPO are outlined in Algorithm 1 in Appendix A.5.

## 3 Experiments

### 3.1 Experimental Setup

**Tasks.** We perform experiments on 6 language understanding tasks and 2 math reasoning tasks to validate our methods, including topic classification (AG's News (Zhang et al., 2015)), natural language inference (BoolQ (Clark et al., 2019)), sentiment classification (IMDB (Maas et al., 2011), TweetEval Emotion (Mohammad et al., 2018)), multi-choice QA (CosmosQA (Huang et al., 2019), HellaSwag (Zellers et al., 2019)), and math reasoning (GSM8K (Cobbe et al., 2021), SVAMP (Patel et al., 2021)). These tasks are widely studied in prompting settings, and hence many expert-crafted and machine-generated prompts are available, which facilitates our offline data collection procedure.

**Baselines.** We compare our method with three types of baselines, including manual prompt engineering (PromptSource (Bach et al., 2022), Chain-of-Thought (Wei et al., 2022)), online prompt optimization (Low Perplexity (Gonen et al., 2022), RLPrompt (Deng et al., 2022), APE (Zhou et al., 2022)) and offline prompting approach (Prompt-OIRL (Sun et al., 2023)). We also compare the prompts rewritten by Chat-GPT. We reproduce the online methods based on InstructEval (Ajith et al., 2023). Rather than making a direct while unfair comparison between the performance of QPO and newer online algorithms, our primary focus is to demonstrate that QPO can further enhance performance by optimizing prompts at query-level based on the prompts obtained through online optimization. Notably, for fair comparison, we use a larger dataset in Prompt-OIRL than ours after final loop data augmentation to eliminate our advances in additional interactions.

**LLMs.** Our method is model-agnostic for policy model, and we use GPT-2 (Radford et al., 2019) in this paper, which is compact while has sufficient text generation capability. For the target LLMs, we use publicly available Llama2-7b-chat (Touvron et al., 2023) for natural language understanding tasks, while for the more challenging math reasoning tasks, we opt for GPT-3.5-turbo (OpenAI, 2023) and GPT-4o (OpenAI, 2024). Moreover, to evaluate the cross-model generalization ability of our trained policy model, we employ models at different abilities, scaling from the GPTNeo-1.3b (Black et al., 2021) to Llama2-13b-chat (Touvron et al., 2023) and Vicuna-13b (Zheng et al., 2024).

**Implementation Details.** For the initial data collection, we utilize 30 expert prompts obtained from InstructEval and 120 prompts rewritten from ChatGPT-3.5 to construct the dataset. We set the QPO with 3 loops. For all the tasks, we use a unified hyperparameter set and do not need complex hyperparameter design for specific task. We do not cherry-pick checkpoints and directly use the final checkpoint in each loop for evaluation and next loop's training. For NLU tasks, we evaluate our method on both zero-shot and few-shot settings, while for math reasoning tasks for GPT-3.5 and GPT-4o, we only test its zero-shot performance due to budget limitation. Both training and testing are conducted on 3 seeds. We set the maximum expected reward as 100 and pick the model with the highest score on the development set and report its score on the test set. For data augmentation, we adopt sampling generation with a top-k of 2 and top-p of 0.9, while for evaluation, we adopt greedy generation. Few-shot evaluations are conducted separately for 3-shot and 6-shot, where the few-shot demonstrations are randomly sampled from the training set of the

Table 1: Main results (accuracy) of QPO and baselines on Llama2 on NLU tasks.

| Method | AG News | BoolQ | IMDB | Emotion | CosmosQA | HellaSwag | Avg. |
|---|---|---|---|---|---|---|---|
| Zero-shot Accuracy | | | | | | | |
| PromptSource | 68.3 | 59.0 | 86.3 | 57.3 | 42.9 | 68.8 | 63.8 |
| ChatGPT | 64.2(0.84) | 62.0(2.41) | 85.0(2.19) | 62.4(2.68) | 42.6(0.51) | 68.8(0.51) | 64.2 |
| Low Perplexity | 70.3(1.04) | 61.0(2.65) | 78.7(2.21) | 57.6(1.71) | 42.0(0.67) | 69.6(0.69) | 63.2 |
| RLPrompt | 47.8(4.67) | 64.9(2.27) | 86.4(2.04) | 49.8(1.17) | 44.7(0.33) | 68.6(0.19) | 60.4 |
| APE | 66.5(1.12) | 64.4(1.50) | 77.0(3.84) | 58.3(2.73) | 43.2(1.17) | 68.3(0.84) | 62.9 |
| Prompt-OIRL | 78.3 | 65.0 | 89.0 | 68.3 | 42.3 | 68.3 | 68.5 |
| QPO | **79.0**(0.44) | **71.0**(1.02) | **91.2**(2.12) | **68.7**(1.07) | **45.3**(0.42) | **69.7**(0.71) | **70.9** |
| Few-shot Accuracy | | | | | | | |
| PromptSource | 83.3 | 67.1 | 89.4 | 66.4 | 45.6 | 70.3 | 70.4 |
| ChatGPT | 83.8(0.75) | 68.1(0.39) | 89.2(0.25) | 67.2(1.78) | 45.5(0.62) | 69.8(0.52) | 70.6 |
| Low Perplexity | 82.9(0.20) | 68.5(0.47) | 89.1(0.09) | 67.6(1.49) | 45.4(0.41) | 70.4(0.13) | 70.7 |
| RLPrompt | 82.8(0.24) | 71.2(0.50) | 89.1(0.54) | 65.4(0.70) | 46.2(0.32) | 70.2(0.57) | 70.8 |
| APE | 83.1(0.33) | 68.4(0.79) | 89.2(0.45) | 67.9(0.58) | 45.8(0.70) | 70.2(0.12) | 70.8 |
| Prompt-OIRL | 83.8 | 73.3 | 89.5 | 67.6 | 48.1 | 70.4 | 72.1 |
| QPO | **85.6**(0.29) | **76.5**(0.62) | **91.4**(0.16) | **69.1**(1.13) | **51.2**(0.41) | **71.0**(0.29) | **74.2** |

Table 2: Main results (accuracy) of QPO and baselines on GPT-3.5 on math reasoning (MR) tasks.

| Method | GSM8K | SVAMP | Avg. |
|---|---|---|---|
| CoT | 88.0(0.49) | 81.0(0.52) | 84.5 |
| ChatGPT | 82.1(1.07) | 79.5(2.24) | 80.8 |
| APE | 85.8(0.74) | 81.3(1.62) | 83.6 |
| Prompt-OIRL | 87.5 | 83.7 | 85.6 |
| QPO | **90.3**(0.68) | **86.8**(0.41) | **88.6** |

tasks. We measure the algorithm's performance using accuracy, defined as the ratio of correctly answered queries by the LLM to the total number of tested queries. More details are demonstrated in Appendix B.1. The code is available at here.

## 3.2 Main Results

**Natural Understanding tasks.** Table 1 presents a comprehensive comparison of query-dependent optimized prompts generated by QPO against human prompts, query-agnostic prompt optimization methods, and SOTA offline query-dependent prompt selection method on Llama2-7b-chat on zero-shot and 6-shot settings. The results show that: **(1)** QPO delivers significantly better results on zero-shot metric (avg. +7.2% over other baselines) and also outperforms other baselines on few-shot setting (avg. +3.3%), where the contextual demonstrations slightly diminish the critical role of prompts, as evidenced by the standard deviation in few-shot scenarios being significantly smaller than in zero-shot scenarios. The 3-shot results in Table 11 further illustrate that our algorithm's advantage becomes more pronounced as the information provided by contextual demonstrations decreases (avg. +6.1%). **(2)** Based on the dataset constructed from these online algorithms, both QPO and Prompt-OIRL exhibit better performance to those optimized at the task level on all three settings (avg. +7.2%, +2.3%, +4.9%, respectively), which underscores the effectiveness of optimizing prompts at the query level. **(3)** Despite the comparative strength of Prompt-OIRL as a discriminative model over our generative model, our algorithm outperforms Prompt-OIRL, demonstrating that learning to generate optimal prompts for specific queries is superior to merely selecting the most suitable ones.

**Math Reasoning tasks.** The experimental results of GPT-3.5 on zero-shot math reasoning tasks are presented in Table 2. When applied to more powerful models for solving complex tasks, QPO exhibits the same leading capabilities. Specifically, QPO achieves up to 7.8% improvement with an average of 5.0% over other baselines, which demonstrates that QPO is a good method for prompting fancy LLMs for these challenging tasks. The results of GPT-4o are shown in Appendix C.2.

### 3.3 Analysis

**Multi-loop augmentation can bootstrap to improve performance in a interaction-cost-efficient way.** Multi-Loop Augmentation (MLA) in our method is supposed to improve the exploration of the query space and prompt space efficiently, so that the augmented dataset can enhance the policy model in turn. The average number of queries in datasets increases from 283 in original dataset to 673 in the final augmented dataset, and the average number of prompts rises from 150 to 829, which obtains 138% and 453% increases in question coverage and prompt diversity with only a 17.1% increase in total data volume. We limit the amount of data augmented in each loop, as research (Wang et al., 2024c) indicates that generated data comprising 10% of the real data provides the maximum training benefit, whereas excessive generated data can be counterproductive. To further quantify the effect of MLA, we conduct ablation

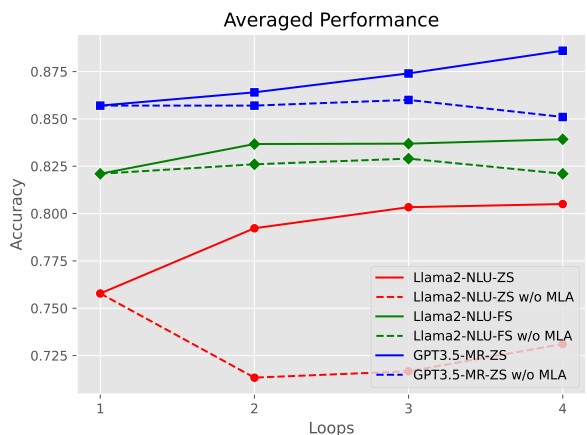

Figure 3: Comparison of QPO and QPO w/o MLA.

experiments that finetune the model the same iterations without data augmentation. As shown in Figure 3, in 3 NLU tasks (AG News, BoolQ, IMDB) and 2 math reasoning (MR) tasks (GSM8K and SVAMP), **MLA can consistently improve the performance in all settings**, while training with more epochs without data augmentation results in significant drops due to overfitting. To be specific, multi-loop augmentation and training lead to a performance improvement of 6.7% and 2.3% for Llama2 in zero-shot and few-shot settings in NLU tasks, respectively, while in the zero-shot math reasoning tasks for GPT-3.5, the performance improved by 3.5%.

Despite the introduction of online LLM interactions in MLA, our method: (1) only engages new interactions to supplement the dataset after the training convergence in each loop, rather than requiring interactions at each step to provide real-time guidance for optimization direction; (2) requires significantly fewer interactions compared to other online methods. Therefore, QPO is considered an offline approach. To demonstrate this, we compared our method with the classic online algorithm APE, which

Table 3: Interaction costs of QPO and APE on Llama2-7b-chat on AG News.

| Method | Model Cost (GPU Hour) |
|--------|----------------------|
| APE | 1.761 |
| Ours | 0.291 |

is a paradigm for many online prompt optimization methods. Table 3 presents the LLM inference time costs, which is operated locally on NVIDIA V100 GPU. To be specific, for 4 loops of training, 3 data augmentation processes are employed, resulting in each MLA requiring approximately 0.097h of computational resources. **Given an average performance improvement of around 2% per loop, this expense for interaction is relatively justified and valuable.** Thus, employing QPO for prompt optimization is substantially more cost-efficient than methods reliant on LLMs as critics. We analyse the specific number of required interactions for these methods in Appendix C.3.

**Reward Matters.** To validate the effects of RL and our proposed reward prediction loss, we conduct ablation experiments comparing supervised fine-tuning (SFT), RL, and QPO (RL+Reward Loss). Table 4 presents that **RL+Reward Loss consistently surpasses others in all evaluation tasks**, which confirms the effects of the additional reward loss. It instruct the model focus more on the rewards, enabling it to determine which query-prompt pairs can achieve higher rewards, allowing the policy model to generate optimal

Table 4: Comparison between SFT, RL and QPO (RL+Reward Loss).

| Method | AG News | | BoolQ | | IMDB | | Avg. | |
|--------|---------|---------|-------|---------|------|---------|------|---------|
| | zero-shot | few-shot | zero-shot | few-shot | zero-shot | few-shot | zero-shot | few-shot |
| SFT | 76.0 | 84.2 | 70.7 | 74.9 | 90.7 | 89.6 | 79.1 | 82.9 |
| RL | 78.7 | 84.3 | 65.0 | 75.2 | 91.0 | 89.6 | 78.2 | 83.1 |
| QPO | **79.0** | **85.6** | **71.0** | **76.5** | **91.2** | **91.4** | **80.4** | **84.5** |

prompts given the maximum expected reward. Compared with SFT, RL also achieves better performance (5/6) because of the high data utilization, which is also thanks to the introduction of the reward. Detailed reason is analysed in Appendix A.1. The reason for the drop between QPO and RL can be attributed to the underestimation of rewards. As the reward and each query token have equal weight contributing to the prompt generation, excessively long queries may cause the model to overlook the single-token reward. Notably, SFT also yields strong results, indicating that fine-tuning PLMs to generate optimal prompts can serve as a general paradigm, regardless of the specific fine-tuning method. The success of SFT also implies that each query does not necessarily need multiple prompts in the dataset; as our data augmentation approach, to balance interaction frequency with query exploration, each new query might only be accessed minimal times, resulting in a limited number of different corresponding prompts generated by sampling. Such augmented data can still positively contribute to the training process. For more details on ablation experiments related to reward design, including the contributions of $\mathcal{R}_{query}$ and $\mathcal{R}_{task}$, the importance of fine-grained perplexity penalty in $\mathcal{R}_{query}$, and the difference between reward predicting positions, please refer to Appendix C.4.

Table 5: Performance comparison between Nearest Neighbor and QPO.

| Method | AG News | | BoolQ | | IMDB | | Avg. | |
|--------|---------|---------|-------|---------|------|---------|------|---------|
| | zero-shot | few-shot | zero-shot | few-shot | zero-shot | few-shot | zero-shot | few-shot |
| NN | 70.3 | 82.4 | 64.3 | 76.7 | 83.7 | 89.1 | 72.8 | 82.7 |
| QPO | **79.0** | **85.6** | **71.0** | **76.5** | **91.2** | **91.4** | **80.4** | **84.5** |

**Ablation on Prompt Generation.** To further validate the rationality of using transformer for query encoding and prompt generation simultaneously, we additionally employ a direct query-dependent baseline for comparison. Specifically, we ablate the text generation component of the policy model, only utilize it to encode queries into query embeddings, select the training query most similar to the testing query using Nearest Neighbor (NN) criterion, and directly choose the prompt with the highest reward associated with the matched training query. The results are depicted in Table 5. Generating specific prompts (QPO) significantly outperforms merely selecting suitable ones (NN), indicating the effectiveness of the main training objective. Notably, NN also achieves higher scores than those query-agnostic prompts shown in Table 1, which illustrates that the trained policy model can accurately obtain the input query embeddings, even without the specific training objective for encoding. And query-specific prompts, whether selected or generated, indeed outperform query-agnostic prompts.

**Ablation on Dataset.** We investigate the effect of initial prompt quantity and overall data quality in the offline dataset on QPO. Firstly, we compare the performance with different numbers of prompts shown in Table 6 , where the 30 prompts are from online algorithms without ChatGPT rewriting and the 5 prompts are randomly selected from the 30 prompts. The results reveal an overall trend that **more prompts lead to better final performance**, while an exceptionally large number of initial prompts is not necessary, as **a moderate amount can already yield good results**. This is because once the dataset contains a sufficient number of high-quality prompts, subsequent data augmentation phase can also effectively enhance prompt diversity.

Table 6: Performance under different prompt availability. "# prompts" demonstrates the number of initial prompts used for training.

| # prompt | Avg. | |
|----------|------|---------|
| | zero-shot | few-shot |
| scarce (5) | 72.8 | 83.1 |
| middle (30) | 77.8 | 84.2 |
| rich (150) | **80.4** | **84.5** |

It also validates the effectiveness of supplementing prompts with lower-quality GPT-rewritten prompts, as a prompt with a lower average accuracy can still perform well on specific queries. This experiment demonstrates QPO's potential to perform well with a smaller number of prompts. Detailed results are shown in Table 16.

Next, we discuss the impact of data quality on our method to illustrate the importance of data filtering. We view the filtered dataset as expert dataset and the original dataset as medium-expert one, for all the prompts within the dataset are capable of achieving at least moderate performances. Table 7 demonstrates that QPO on expert dataset after data filtering outperforms that on medium-expert dataset. By filtering 34.4% of the lower-quality data, our method achieves a 2.1% performance improvement. While when trained on an unfiltered medium-expert dataset, our algorithm still achieves comparable or better results to the SOTA Prompt-OIRL, which demonstrates that **QPO exhibits strong robustness to prompt quality and data filtering can further enhance its performance**. The detailed results are presented in Table 18.

Table 7: Performance under different data quality. QPO with medium-expert dataset can already achieve comparable or better results to SOTA Prompt-OIRL.

| Dataset | Avg. | |
|---|---|---|
| | zero-shot | few-shot |
| expert | **80.4** | **84.5** |
| medium-expert | 77.9 | 82.6 |

Table 8: Cross-model generalization on different models.

| Model | Method | AG News | | BoolQ | | IMDB | | # wins |
|---|---|---|---|---|---|---|---|---|
| | | zero-shot | few-shot | zero-shot | few-shot | zero-shot | few-shot | |
| GPTNeo-1.3b | PromptSource | 58.5 | 63.0 | **50.3** | **49.8** | 80.9 | 82.9 | 2 |
| | Prompt-OIRL | **65.3** | **65.1** | 48.3 | 49.0 | 78.3 | 84.0 | 2 |
| | **QPO** | 61.7 | 63.4 | **50.3** | 49.1 | **83.3** | **87.5** | 3 |
| LlaMa2-13b-chat | PromptSource | 75.1 | 83.7 | 79.1 | 82.8 | **89.8** | **92.2** | 1 |
| | Prompt-OIRL | **80.3** | 82.4 | 78.0 | 83.0 | 88.3 | 91.2 | 1 |
| | **QPO** | **80.3** | **84.4** | **80.7** | **83.9** | 90.3 | 92.1 | 5 |
| Vicuna-13b | PromptSource | 74.3 | 80.3 | 79.4 | 80.8 | **89.5** | 92.2 | 1 |
| | Prompt-OIRL | **76.0** | 81.4 | 82.7 | 84.1 | 88.3 | 92.5 | 1 |
| | **QPO** | 75.0 | **82.6** | **86.0** | **84.6** | 87.3 | **92.9** | 4 |

**Cross-Model Generalization.** We investigate whether QPO can be used to steer the target LLMs which are not involved in the dataset collection and augmentation. In this section, the policy model are trained on datasets collected by Llama2-7b-chat. As shown in Table 8, QPO demonstrates excellent cross-model transferability compared with human designed prompts, which are generally considered high-quality and can be employed on any LLMs. Compared to the results with GPTNeo-1.3B, our algorithm demonstrates greater advantages on two larger LLMs (3 wins v.s. 5 wins & 4 wins). This cross-model generalization significantly enhances the value of our approach: We can train the policy model using existing data tested on different models and then apply it to the target model. Additionally, we can collect data using a cost-effective smaller LLM and train the policy model, then directly apply it to a more expensive larger LLM.

**Case Study.** The prompts generated by QPO in NLU tasks are presented in Table 19, which showcase the versatility of our approach. Notably, while their individual accuracies on all testing questions are relatively low, their collective application significantly enhances task-level performance when aligned with specific queries. **Although a single generated prompt may not be suitable for all queries, it is certainly the best for the specific query.**

## 4 Related Works

**Prompt Optimization.** Automatic prompt optimization has become a pivotal issue in domain of LLMs. Recently, there has been a growing interest in this area. Soft prompts (Li & Liang, 2021; Zhang et al., 2021;

Lester et al., 2021; Liu et al., 2023b) achieve effective performance by tuning the parameters of the input tokens, while exhibit two significant drawbacks. They require parameters of the target LLM, which are inaccessible for black-box APIs, and their vector representation lacks interpretability for humans. Discrete prompts, also known as prompt engineering, eliminate the shortcomings and also achieve great success. Most of these methods are query-agnostic. Automatic Prompt Engineering (APE)(Zhou et al., 2022) induces prompts by instructing a LLM to describe the given task and refines the set of generated prompts. Low Perplexity (SPELL)(Gonen et al., 2022) finds the perplexity is negatively correlated with the performance and select the least perplexity instructions. RLPrompt (Deng et al., 2022) employs RL to optimize a prompt policy. Recently, more studies focus on leveraging LLM as prompt optimizer using black-box optimization method for swift prompt engineering, such as evolution algorithm (Guo et al., 2023; Fernando et al., 2023), alternate sampling (Zhou et al., 2022; Xu et al., 2023; Pryzant et al., 2023; Kong et al., 2023), and RL (Kong et al., 2024; Wang et al., 2023b). However, studies (Wu et al., 2022; Jiang et al., 2022; Zhang et al., 2022; Sun et al., 2023) have shown that query-dependent prompts can obtain better results. TEMPERA (Zhang et al., 2022) designs the action space to be editing operations to achieve test-time query-dependent edit. Prompt-OIRL (Sun et al., 2023) utilize inverse RL to learn a proxy reward model and thus select the best prompt based on the query from a candidate prompt set. Another advantage of Prompt-OIRL is it makes full use of offline dataset, significantly reducing the inference cost by interactions with LLMs. Promptist (Hao et al., 2023) also collect an offline dataset and employ SFT to fine-tune the policy model, while the main performance improvements are from the subsequent online RL.

**Learning from Human Feedback.** Our method is related to the research on reinforcement learning from human feedback, which has shown promising results on a wide range of tasks, including NLP (Rajpurkar et al., 2018; Wang et al., 2023a; Ouyang et al., 2022; Stiennon et al., 2020; Klie et al., 2020; Wang et al., 2024a) and classical RL (Christiano et al., 2017; Ibarz et al., 2018; Xia et al., 2024). Another similar category is reinforcement learning form AI feedback (RLAIF) (Bai et al., 2022; Lee et al., 2023), which leverages the feedback typically from LLMs to facilitate RL. Differently, we directly use the feedback from LLM to shape rewards rather than learning a reward model. Our method is akin to imitation learning (Torabi et al., 2018; Chen et al., 2021; Janner et al., 2021), directly learning a policy from a batch of behavioral demonstrations and their corresponding rewards. This distinctive approach sets our method apart from existing prompt optimization techniques.

## 5 Conclusion and Future Work

We propose QPO, a novel query-dependent prompt optimization method through a multi-loop offline RL framework, which is cost-efficient. This method leverages offline datasets from existing evaluations and employs offline RL to fine-tune a small-scale language model as the policy model, generating query-level optimal prompts. Experimental results demonstrate the superiority and cross-model generalization ability of QPO, significantly enhancing the value of our method. Besides, we conduct extensive ablation studies to analyse our method's insights and rationale. While our method has substantially reduced the required online interactions, we can entirely eliminate these interactions if necessary. For instance, in certain business scenarios where only a fixed amount of reward-labeled data is available and the reward labeling rules are unknown, it is impossible to use LLMs to obtain new labeled data as our exploration. We can employ inverse reinforcement learning to derive a reward model from the original dataset to score the new query-prompt pairs obtained through the multiple-loop data augmentation. This approach completely eliminates the need for online interactions with LLMs, which we consider for future work. Furthermore, in the domains of text-to-image and text-to-video generation, where LLMs have slower inference speeds, interactions are more expensive, and user queries are more unique, our offline query-dependent prompt optimization methods will show greater potential and advantages.

## 6 Broader Impact Statements

This paper presents work whose goal is to advance the field of Machine Learning. There are many potential societal consequences of our work, none which we feel must be specifically highlighted here.

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

# A Extended Discussion on QPO

## A.1 Advantages of offline RL compared to SFT

Supervised fine-tuning (SFT) can be regarded as a form of imitation learning (Zheng et al., 2022), similar to behavior cloning (Torabi et al., 2018), where it mimics expert actions (i.e., the query-specific optimal prompts) based on given states (i.e., input questions). While offline reinforcement learning introduces reward values, guiding the generation of prompts based on the given queries and associated rewards. In prompt generation task, directly employing SFT results in low data utilization because it is unknown in advance that which prompt is optimal for specific problems. This necessitates massive prior evaluation of all prompts before selecting the best one as the expert behavior to construct the dataset, leading to significant resource wastage. Specifically, in the final SFT dataset, each query should ideally have only one prompt label. All $M$ prompts must be evaluated but only one optimal prompt is retained as the label for SFT ultimately, wasting the remaining $(M-1)/M$ evaluation data. While for offline RL, the prediction of prompt is based on both reward and query, leading to a much more diversity in input. Since different prompts have various rewards, most of them can be utilized for training. Offline RL offers significant advantages through the introduction of the rewards, which allows each query to be associated with multiple prompts and their corresponding rewards in the dataset. On one hand, the rewards improves the utilization of collected suboptimal prompts for training, significantly boosting exploration of the prompt space while avoiding resource wastage caused by prompt selection. On the other hand, rewards in RL enable the policy model to distinguish the prompting abilities of different prompts for various queries at the token level, leading to the generation of novel prompts for unknown queries.

## A.2 Prompt Generation as Single-step Decision Process

In the reinforcement learning setup, prompt generation is a sparse reward task. The complete prompt is evaluated with the target LLM to obtain a reward value only after it is fully generated. Given the availability of substantial amounts of such test data, offline reinforcement learning is an effective approach for prompt optimization. For offline RL, there are two methods for collecting token-wise rewards which can be used for multi-step decision-making processes:

(1) Manually designing token-level rewards, which not only requires extensive prior expert knowledge but also introduces additional errors, making it unreliable.

(2) Iteratively deleting the prompt's tokens from end to beginning and re-evaluating to obtain rewards from $s_{m-1} = (\boldsymbol{q}, p_1, ..., p_{m-1})$ to $s_m = (\boldsymbol{q}, p_1, ..., p_m)$, where $p_m$ denotes the $m$-th token in the complete prompt. However, this approach significantly increases the cost of collecting datasets and is thus impractical.

Therefore, we do not introduce additional proxy rewards and directly utilize the sparse rewards. At this point, instead of modeling the prompt generation task as a multi-step decision-making process $\{r, s_0, p_1, r, s_1, p_2, ...\}$, where each $r$ is the same sparse reward and additional methods are required to encode long texts $(q_1, ..., q_n, p1, ..., p_i)$ into the single token state $s_i$, we simplify the task modeling based on the principle of Occam's Razor. Thus, the prompt generation is formulated into a single-step decision process $(r, \boldsymbol{q}, \boldsymbol{p})$. Notably, the single-step decision-making process refers to the reward and state, taking only one action (i.e., generating the prompt), while the inner prompt generation process remains an autoregressive prediction $\boldsymbol{p} \sim \prod_t \pi(p_t | r, \boldsymbol{q}, \boldsymbol{p}_{<t})$. The single-step decision process also resembles the offline contextual bandit setup, as actions do not interact with the environment.

## A.3 The Architecture of Policy Model

Compared to the standard GPT-2 model with 124M parameters, we make three modifications to the model architecture: a reward embedding layer, a reward prediction layer, and a timestep embedding layer. Both the reward embedding layer and reward prediction layer are single-layer MLPs at the first token of the policy model. The reward is encoder into a reward encoding through the reward embedding layer, then input as the first token into the transformer to obtain the output at the first token position, which is then passed through

the reward prediction layer to generate the predicted reward. This process can be viewed as an autoencoder designed to extract more information from the reward. The additional timestep embedding layer encoding three timesteps into three embeddings, which are added with reward embedding, query token embedding, and prompt token embedding, respectively. This helps the policy model better learn the positional relationships and distinctions between reward, question and prompt.

### A.4 Autoencoder with reward prediction loss

Autoencoder encodes the input into a low-dimensional latent state through an encoder and subsequently learns to reconstruct the original input from this low-dimensional latent state via a decoder. In this process, the latent state can effectively represent the original input. In our approach, the reward embedding layer and the transformer map the reward value into a hidden state. During training, the additionally introduced reward prediction layer acts as a decoder to predict the original input reward based on this hidden state, and the encoder-decoder is updated through the reward prediction loss. In the inference phase, we no longer utilize the reward prediction layer (decoder); instead, we leverage the hidden state (related to the reward) obtained from the encoder to perform another task, that is, predicting the prompt. This process mirrors the autoencoder's principle, thus introducing the reward prediction loss and the reward prediction layer into the PLM essentially makes it serve as an autoencoder. Notably, the reward prediction loss does not directly contribute to reinforcement learning process, as its optimization objective is not to generate actions.

### A.5 Detailed Algorithm

---
**Algorithm 1** Query-Dependent Prompt Optimization

---
**Require:** Initial dataset $\mathcal{D}_0$, a collection set for task queries $\mathcal{Q}$, the number of loops $T$, a pre-trained language model $\pi_0$, ORL$(\cdot)$ denotes the offline RL fine-tuning process, $\mathcal{R}(\cdot)$ presents evaluating the new query-prompt pairs on the target LLM and calculating the reward.

1: **Offline RL:** $\pi_1 \leftarrow \text{ORL}(\pi_0, \mathcal{D}_0)$
2: **for** $t = 1$ to $T - 1$ **do**
3:      **Query Augmentation:** $q \leftarrow \text{random\_sample}(\mathcal{Q})$
4:      **Prompt Augmentation:** $p \leftarrow \pi_t(q)$
5:      **Dataset Augmentation:** $\mathcal{D}_A \leftarrow \{q, p, \mathcal{R}(q, p)\}$
6:                      $\mathcal{D}_t \leftarrow \{\mathcal{D}_{t-1}, \mathcal{D}_A\}$
7:      **Offline RL:** $\pi_{t+1} \leftarrow \text{ORL}(\pi_t, \mathcal{D}_t)$
8: **end for**
9: **return** the policy model $\pi_T$, which can generate optimal query-dependent prompts during testing.

---

## B Supplemental Experiment Detials

### B.1 Hyperparameters and Details

For the task datasets with default testing or development set, we use their original split to obtain our testing set. If there is no official training/development/testing split, we randomly sample a reasonably large set for stable evaluating and testing. Additionally, for all tasks, we split 10% training samples as collection set for initial query collection and subsequent query augmentation, ensuring that the collected queries for training the policy model do not appear in the in-context examples during few-shot evaluation, and simultaneously simulating a scenario where very few questions are available. In each loop in query augmentation, 1,000 queries are sampled, which may include repeated queries. For repeated queries, the sampling prompt generation approach ensures different prompts are generated, allowing for the exploration of varying prompt effectiveness for each query. We do not explicitly aim for uniqueness in the sampled 1,000 queries per loop, opting instead for a fully random sampling approach.

The parameters for the experiments are shown in Table 9. Notably, QPO consists of 4 loops, indicating there are 4 times of offline reinforcement learning fine-tuning stage and 3 times of data augmentation between

each fine-tuning stage. Since we introduce additional networks into the policy model, the number of training epochs and the learning rate required for the first loop of training are relatively high.

Table 9: Hyperparameter settings of QPO

| Hyperparameters | Values |
|---|---|
| Loops | 4 |
| Batchsize | 128 |
| Learning Rate | 1e-3 for the 1st loop, 1e-4 for others |
| Train Epochs | 100 for the 1st loop, 20 for others |
| Optimizer | AdamW |
| Weight Decay | 1e-4 |
| Balancing Parameter $\lambda$ | 0.1 |

## B.2  Offline Dataset Collection

For the initial dataset collection, testing 150 prompts on too many queries incurs excessive computational costs. Therefore, we test 100 queries for every group of 40 prompts (with the last group containing 30 prompts). Each group sampled different queries, achieving a balance between query coverage and computational cost. Since queries are randomly sampled for each test group, there are overlapped queries across different groups. In the end, our dataset consists of 15,000 paired examples, comprising 150 prompts and approximately 350 queries. This dataset size is significantly lower than that required by other offline algorithms, such as Prompt-OIRL for averages 56,900 examples and Promptist for averages 360,000 examples.

For the specific prompts, the initial 30 prompts are derived from various previous approaches, including PromptSource, ChatGPT Rewrite, Low Perplexity, RLPrompt and APE, which are all produced by InstructEval (Ajith et al., 2023), and we leverage ChatGPT-3.5 chat box to rewrite 120 new prompts based on these 30 prompts through in-context learning. Specifically, we provide 5 existing prompts to ChatGPT-3.5, and ask it to generate other 30 effective prompts for 4 times. The prompt we use for rewriting is demonstrated in Table 10.

Table 10: Template used for ChatGPT rewriting prompts (Sun et al., 2023)).

In practice, people find the following prompts help improving the classification abilities of language models like LlaMa-7b-chat on the <TASK> dataset. Rewrite the following instructions via rephrasing and/or adding specific requirements. Use illustrative description if needed.

#########
Prompts:
<PROMPT1>
<PROMPT2>
<PROMPT3>
<PROMPT4>
<PROMPT5>

Please provide 30 more prompts different from those prompts that may improve the classification ability of language models. Diversity is much appriciated!

## B.3  Codes and Hardware

Our code, as well as the offline datasets, is now publicly available at here. All experiments are conducted on a single NVIDIA V100 32g GPU.

# C  Additional Results

## C.1  Main Results on 3-shot Evaluation

Table 11: Main results of QPO and baselines on NLU tasks on 3-shot settings

| Method | AG News | BoolQ | IMDB | Emotion | CosmosQA | HellaSwag | Avg. |
|---|---|---|---|---|---|---|---|
| PromptSource | 76.0 | 58.5 | 90.3 | 42.2 | 39.3 | 69.8 | 62.7 |
| Low Perplexity | 53.5 | 58.0 | 90.1 | 48.0 | 39.0 | 70.0 | 59.8 |
| RLPrompt | 76.2 | 69.4 | 89.7 | 56.1 | 41.1 | 69.9 | 67.1 |
| APE | 58.6 | 59.9 | 89.9 | 57.6 | 40.2 | 70.2 | 62.7 |
| Prompt-OIRL | 76.9 | 67.1 | 89.2 | 52.7 | 40.6 | 70.4 | 66.2 |
| QPO | **82.3** | **71.1** | **90.5** | **62.7** | **41.5** | **70.7** | **69.8** |

## C.2  Main Results on GPT-4o.

As demonstrated in Table 12, our method also outperforms other baselines on most updated models.

## C.3  Cost Analysis

In this section, we analyse the specific number of interactions required by different algorithms. APE requires a total of $M*|D|*T$ interactions with LLMs to obtain the feedback for optimization, while QPO only requires $|C|*T'$ interactions for data augmentation, where $M$ is the candidate population under evaluation, $|D|$ is the size of development set, $|C|$ is the size of collection set, and $T$ and $T'$ are the number of iterations needed. Though $|C|$ is slightly larger than $|D|$ for better exploration, QPO significantly reduces computational overhead due to its minimal iteration $T'$ and the absence of the need for evaluate $M$ candidate prompts.

## C.4  Analysis on reward design

**Both $\mathcal{R}_{query}$ and $\mathcal{R}_{task}$ contribute to the performance.** By separately removing $\mathcal{R}_{query}$ and $\mathcal{R}_{task}$ and compare the results with the full reward design($\mathcal{R}_{query}+\mathcal{R}_{task}$), results in Table 13 indicate that for most of tasks, the combined design of $\mathcal{R}_{query}+\mathcal{R}_{task}$ yields superior performance. Thus, both $\mathcal{R}_{query}$ and $\mathcal{R}_{task}$ contribute to the performance. Notably, the use of either $\mathcal{R}_{query}$ or $\mathcal{R}_{task}$ alone also demonstrates commendable results, which underscores the rationality of employing these two metrics as rewards. Furthermore, the comparison between using $\mathcal{R}_{query}$ and $\mathcal{R}_{task}$ individually highlights the advantage of $\mathcal{R}_{query}$, suggesting that query-based prompt optimization is more effective than task-level prompt optimization.

**Perplexity penalty is important.** We compare the performance of rewards with and without perplexity (PPL) to validate the importance of the fine-grained penalty in $\mathcal{R}_{query}$. The experimental results are presented in Table 14. For all tasks, incorporating PPL consistently yields better performance. This is because the introduction of PPL transforms the $\mathcal{R}_{query}$ from a binary value into a fractional value, enhancing the discrimination among different query-prompt pairs. Consequently, the policy model can better learn which prompts are more effective for specific queries.

**Predicting reward at the first token helps learn better.** We introduce the reward prediction loss to enhance the model's ability to discern the reward values across different query-prompt pairs, while simultaneously preventing the policy model from overemphasizing rewards to the extent that it neglects the distinctions between queries and prompts, potentially leading to overfitting on the prompt with the highest reward value across all samples in the dataset. Predicting the reward at the first token can simplify the attention calculation process and mitigate excessive updates to the query and prompt embedding layers, thereby reducing the risk of overfitting. Our experimental validation in Table 15 confirms the effectiveness of predicting the reward value at the first token, as evidenced by its better performance compared to predicting at the last token.

Table 12: Main results of QPO and baselines on GPT-4o on GSM8K.

| Method | GSM8K |
|---|---|
| CoT | 95.8 |
| ChatGPT | 94.7 |
| APE | 95.0 |
| Prompt-OIRL | 93.5 |
| QPO | **96.5** |

Table 13: Comparisons between $\mathcal{R}_{query}$, $\mathcal{R}_{task}$ and $\mathcal{R}_{query}+\mathcal{R}_{task}$.

| Method | AG News | | BoolQ | | IMDB | | Avg. | |
|---|---|---|---|---|---|---|---|---|
| | zero-shot | few-shot | zero-shot | few-shot | zero-shot | few-shot | zero-shot | few-shot |
| $\mathcal{R}_{query}$ | 78.3 | 84.5 | **71.4** | **77.2** | 89.0 | 90.9 | 79.6 | 84.2 |
| $\mathcal{R}_{task}$ | 76.3 | 81.6 | 71.0 | 75.3 | 84.7 | 91.2 | 77.3 | 82.7 |
| $\mathcal{R}_{query}+\mathcal{R}_{task}$ | **79.0** | **85.6** | 71.0 | 76.5 | **91.2** | **91.4** | **80.4** | **84.5** |

### C.5 Results on Different Prompt Quantity

The detailed results are shown in Table 16

### C.6 Results on Different Prompt Quality

We conduct experiments on the dataset where all prompts were rewritten by ChatGPT to validate the robustness of QPO to prompt quality within the dataset. The results are presented in Table 17. The absence of initial high-quality prompts has minimal impact on QPO's performance, as the method focuses on the score of each query-prompt pair rather than the performance of individual prompts. Consequently, even without expert prompts, there are still other prompts perform well for each specific query.

The detailed results on different query-prompt pair quality is shown in Table 18.

### C.7 Case Study

Table 14: Comparisons between with and without perplexity (PPL).

| Method | AG News | BoolQ | IMDB | Emotion | CosmosQA | HellaSwag | Avg. |
|---|---|---|---|---|---|---|---|
| QPO (w/ PPL) | 79.0 | 71.0 | 91.2 | 68.7 | 45.3 | 69.7 | 70.9 |
| QPO (w/o PPL) | 75.0 | 65.7 | 90.3 | 62.7 | 42.7 | 69.0 | 67.6 |

Table 15: Comparisons between reward predicting positions.

| Method | AG News | | BoolQ | | IMDB | | Avg. | |
|---|---|---|---|---|---|---|---|---|
| | zero-shot | few-shot | zero-shot | few-shot | zero-shot | few-shot | zero-shot | few-shot |
| QPO (first token) | 79.0 | 85.6 | 71.0 | 76.5 | 91.2 | 91.4 | 80.4 | 84.5 |
| QPO (last token) | 79.3 | 81.2 | 69.7 | 76.1 | 89.3 | 90.9 | 79.4 | 82.7 |

Table 16: Performance under different prompt availability.

| Prompt Num | AG News | | BoolQ | | IMDB | | Avg. | |
|---|---|---|---|---|---|---|---|---|
| | zero-shot | few-shot | zero-shot | few-shot | zero-shot | few-shot | zero-shot | few-shot |
| scarce | 65.3 | 83.3 | 65.7 | 76.3 | 87.3 | 89.6 | 72.8 | 83.1 |
| middle | 70.7 | 83.9 | 70.7 | **78.4** | **91.5** | **89.7** | 77.6 | 84.0 |
| rich | **79.0** | **85.6** | **71.0** | 76.5 | 91.2 | **91.4** | **80.4** | **84.5** |

Table 17: Performance under different prompt quality in dataset.

| Datase | AG News | | BoolQ | | IMDB | | Avg. | |
|---|---|---|---|---|---|---|---|---|
| | zero-shot | few-shot | zero-shot | few-shot | zero-shot | few-shot | zero-shot | few-shot |
| QPO | **79.0** | 85.6 | **71.0** | **76.5** | **91.2** | **91.4** | **80.4** | **84.5** |
| only ChatGPT prompt | 74.3 | **85.9** | 67.0 | 76.4 | 90.7 | 90.9 | 77.3 | 84.4 |

Table 18: Performance under different query-prompt pair quality.

| Datase | AG News | | BoolQ | | IMDB | | Avg. | |
|---|---|---|---|---|---|---|---|---|
| | zero-shot | few-shot | zero-shot | few-shot | zero-shot | few-shot | zero-shot | few-shot |
| expert | **79.0** | **85.6** | **71.0** | **76.5** | **91.2** | **91.4** | **80.4** | **84.5** |
| medium-expert | 75.0 | 83.1 | 68.3 | 75.9 | 90.3 | 88.9 | 77.9 | 82.6 |

Table 19: Prompts generated by QPO. "I.Acc." denotes the individual accuracy.

| Generated Query-level Prompt | I. Acc. | QPO Acc. |
|---|---|---|
| **AG News** | | |
| In which part of a newspaper, World News, Sports, Business, or Science and Technology? Innov., Sports.., or., or.,.,., or., | 69.0 | 79.0 |
| Assign this news report to the most suitable newspaper segment: World News, Sports, Business, or Technology. and Sports, or Sports, or | 70.0 | |
| ... | ... | |
| **BoolQ** | | |
| Analyze if the passage validates the true/false claim in the question. | 64.0 | 71.0 |
| Confirm the correctness of a true/false question using the text excerpt....... | 63.0 | |
| ... | ... | |
| **IMDB** | | |
| Decipher the overall sentiment conveyed by the reviewer's critique of the movie. or or negative? | 89.3 | 91.2 |
| Interpret the reviewer's emotional stance towards the film: positive or negative??? | 87.7 | |
| ... | ... | |
| **Emotion** | | |
| receive full credit, choose the emotion that best fits the sentiment expressed in the tweet from the following options: anger, joy, optimism, or sadness. sadness. | 49.7 | 68.7 |
| From the given options, determine the emotion that best captures the essence of the tweet's sentiment: anger, joy, optimism, or sadness... tweet. best | 60.0 | |
| ... | ... | |
| **CosmosQA** | | |
| Analyze the contextual details to select the response that offers the most comprehensive understanding. the. | 43.3 | 45.3 |
| Synthesize the information provided below to formulate a well-supported response to the upcoming question. question. | 42.3 | |
| ... | ... | |
| **HellaSwag** | | |
| Explore the passage's thematic depth and devise an ending that explores those themes further......... | 68.0 | 69.7 |
| EnIm with the passage and them and incorporate them into the ending. engaged. engagedates. their comple their expectations. | 67.7 | |
| ... | ... | |

