# OpenReview forum: "QPO: Query-dependent Prompt Optimization via Multi-Loop Offline Reinforcement Learning"
_TMLR — Accepted by TMLR_

### Review · Reviewer_giNB · 2024-09-11

**Summary Of Contributions:**

The main idea of this work are two-folds:
- Most of prompt engineering approaches focus on single task-level prompts, which might lead to suboptimal performance. The proposed method, Queery-dependent Prompt Optimization (QPO) finetunes a small pre-trained language model (PLM) to generate prompts tailored to different queries in the same task.
- To reduce the usage of LLM for prompt optimization, QPO leverages multi-loop offline reinforcement learning on the dataset initially consisting of query-prompt pairs from previous prompt engineering approaches and bootstrapped with the finetuned PLM between each training loop.

The experimental results show the superiority and cost-efficiency of QPO on several natural language understanding tasks and math reasoning tasks.

**Audience:**

Yes

**Broader Impact Concerns:**

Please add the Broader Impact Statement in the final draft.

**Claims And Evidence:**

Yes

**Requested Changes:**

Overall, the quality of the paper is good. Please address the weaknesses above.
Also, minor questions:
- The authors said that QPO is robust to prompt quality. Does QPO achieve competitive results even without initial high-quality prompts, that is only using the rewritten prompts by ChatGPT or the prompts with the rewards below the threshold (66.7th percentile)?

**Strengths And Weaknesses:**

Strengths
- Overall, the paper is well-written.
- The proposed training objective is well designed and makes sense.
- The main experiments show the superiority of QPO and ablation studies provide interesting findings about query-dependent prompt engineering.

Weaknesses
- There is no in-depth analysis on the training objective: e.g., ablation study to verify that both R_query and R_task contribute to the performance, the importance of perplexity penalty in R_query, comparison with the different implementation of reward prediction loss (predict reward at the last token, not the first token), etc.
- There is a lack of explanation/analysis of the reward prediction loss. Please provide the evidence that the finetuned PLM with the reward prediction loss can serve as an autoencoder.
- There are some missing details in the main paper
  - It is not clear that how the initial 30 prompts are constructed. Are they the results of a single previous approach or various approaches?
  - How many queries are sampled at the query augmentation stage in each loop?
  - Please indicate which methods are human-made, query-agnostic, and offline query-dependent in Table 1.
  - Is the data filtering applied after getting 150 prompts? If not, how many prompts remain after filtering?

---

> ### Author Response · Authors · 2024-11-07
> **Rebuttal by Authors**
>
> We are greatly appreciative of the reviewer's recognition of our work and the thoughtful feedback that will surely turn our paper into a better shape. We offer our responses to address your concerns as follows.
>
> >Q1. ablation study to verify that both $R_{query}$ and $R_{task}$ contribute to the performance
>
> We conduct ablation study by separately removing $R_{query}$ and $R_{task}$ and compare the results with our full reward design ($R_{query}$+$R_{task}$), as shown in the table below. The results indicate that for most of tasks, the combined design of $R_{query}$+$R_{task}$ yields superior performance. Thus, both $R_{query}$ and $R_{task}$ contribute to the performance. Notably, the use of either $R_{query}$ or $R_{task}$ alone also demonstrates commendable results, which underscores the rationality of employing these two metrics as rewards. Furthermore, the comparison between using $R_{query}$ and $R_{task}$ individually highlights the advantage of $R_{query}$, suggesting that query-based prompt optimization is more effective than task-level prompt optimization. This experiment and analysis have been appended to Appendix C.4 of the paper.
>
> | | AG News| | BoolQ| | IMDB| | Avg. | |
> |----------|-----------|------------|------------|------------|------------|------------|------------|------------|
> | | zero-shot| few-shot | zero-shot| few-shot |zero-shot| few-shot |zero-shot| few-shot |
> |$R_{query}$|78.3|84.5|71.4|77.2|89.0|90.9|79.6|84.2|
> |$R_{task}$| 76.3| 81.6| 71.0| 75.3| 84.7| 91.2| 77.3| 82.7|
> |$R_{query}+R_{query}$|79.0|85.6|71.0|76.5|91.2|91.4|80.4|84.5|
>
>
> >Q2. the importance of perplexity penalty in R_query.
>
> We compare the performance of rewards with and without perplexity (PPL) to validate the importance of penalty in $R_{query}$. The experimental results are presented in the table below. For all tasks, incorporating PPL consistently yields better performance. This is because the introduction of PPL transformed the $R_{query}$ from a binary value into a fractional value, enhancing the discrimination among different query-prompt pairs. Consequently, the policy model can better learn which prompts are more effective for specific queries. This experiment and analysis have been appended to Appendix C.4 of the paper.
>
> | | AG News|  BoolQ| IMDB| Emotion| CosmosQA| HellaSwag| Avg. |
> |----------|-----------|------------|------------|------------|------------|------------|------------|
> |QPO (w/ PPL) | 79.0| 71.0| 91.2|68.7|45.3|69.7|70.9|
> |QPO (w/o PPL)|75.0|65.7|90.3|62.7|42.7|69.0|67.6|
>
> >Q3. Comparison with the different implementation of reward prediction loss (predict reward at the last token, not the first token)
>
> We introduce the reward prediction loss to enhance the model's ability to discern the reward values across different query-prompt pairs, while simultaneously preventing the policy model from overemphasizing rewards to the extent that it neglects the distinctions between queries and prompts, which may lead to overfitting on the prompt with the highest reward value across all samples in the dataset. Predicting the reward at the first token can simplify the attention calculation process and mitigate excessive updates to the query and prompt embedding layers, thereby reducing the risk of overfitting. Our experimental validation confirms the effectiveness of predicting the reward value at the first token, as evidenced by its better performance compared to predicting at the last token. This experiment and analysis have been appended to Appendix C.4 of the paper.
>
> | | AG News| | BoolQ| | IMDB| | Avg. | |
> |----------|-----------|------------|------------|------------|------------|------------|------------|------------|
> | | zero-shot| few-shot | zero-shot| few-shot |zero-shot| few-shot |zero-shot| few-shot |
> |QPO (first token)|79.0|85.6|71.0|76.5|91.2|91.4|80.4|84.5|
> |QPO (last token)| 79.3| 81.2|69.7|76.1|89.3|90.9|79.4|82.7|

---

> ### Author Response · Authors · 2024-11-07
> **Rebuttal by Authors**
>
> >Q4. There is a lack of explanation/analysis of the reward prediction loss. Please provide the evidence that the finetuned PLM with the reward prediction loss can serve as an autoencoder.
>
> Autoencoders encode the input into a low-dimensional latent state through an encoder and subsequently learn to reconstruct the original input from this low-dimensional latent state via a decoder. In this process, the latent state can effectively represent the original input. In our approach, the reward embedding layer and the transformer map the reward value into a hidden state. During training, the additionally introduced reward prediction layer acts as a decoder to predict the original input reward based on this hidden state, and the encoder-decoder is updated through the reward prediction loss. In the inference phase, we no longer utilize the reward prediction layer (decoder); instead, we leverage the hidden state (related to the reward) obtained from the encoder to perform another task, that is, predicting the prompt. This process mirrors the autoencoder's principle, thus introducing the reward prediction loss and the reward prediction layer into the PLM essentially makes it serve as an autoencoder. This explanation have been appended to Appendix A.4 of the paper.
>
> >Q5. It is not clear that how the initial 30 prompts are constructed. Are they the results of a single previous approach or various approaches?
>
> The initial 30 prompts are derived from various previous approaches, including PromptSource, ChatGPT Rewrite, Low Perplexity, RLPrompt and APE, which are all produced by InstructEval. I updated the explanation of these 30 initial prompts in Appendix B.2 in the paper.
>
> >Q6. How many queries are sampled at the query augmentation stage in each loop?
>
> In each loop, 1,000 queries are sampled, which may include repeated queries. For repeated queries, the sampling prompt generation approach ensures different prompts are generated, allowing for the exploration of varying prompt effectiveness for each query. We do not explicitly aim for uniqueness in the sampled 1,000 queries per loop, opting instead for a fully random sampling approach. I have updated the detail in Appendix B.1.
>
> >Q7.Please indicate which methods are human-made, query-agnostic, and offline query-dependent in Table 1.
>
> PromptSource is human-made prompt engineering approach;  Low Perplexity, RLPrompt, and APE are query-agnostic method; Prompt-OIRL is offline query-dependent method.
> Due to paper space constraints, it’s impractical to specify the category of each method within Table 1. However, we have already detailed the category of each method in paragraph Baseline in the Experimental Setup section; you can refer to that paragraph for more information.
>
> >Q8.Is the data filtering applied after getting 150 prompts? If not, how many prompts remain after filtering?
>
> Yes, the data filtering is applied after getting 150 prompts, and all the 150 prompts remain after filtering. What we filter out are the query-prompt pairs with low scores, rather than poorly performing prompts. Even a prompt with poor average performance can still guide the LLM to answer correctly on specific problems. In such cases, the query-prompt pair is preserved, so the prompt is still in the dataset. Therefore, all prompts are preserved. This detail have been appended to the Data Filtering Paragraph of the paper.
>
> >Q9.Does QPO achieve competitive results even without initial high-quality prompts, that is only using the rewritten prompts by ChatGPT or the prompts with the rewards below the threshold (66.7th percentile)?
>
> We conduct experiments on the dataset where all prompts were rewritten by ChatGPT to validate the robustness of QPO to prompt quality within the dataset. The results are presented in the table below. The absence of initial high-quality prompts has minimal impact on QPO's performance, as the method focuses on the score of each query-prompt pair rather than the performance of individual prompts. Consequently, even without expert prompts, there are still other prompts perform well for each specific query. This experiment and analysis is appended in Appendix C.6.
>
> Due to the limited number of samples below the threshold and the small size of the corresponding dataset, we did not perform the experiment "only using prompts with the rewards below the threshold".
>
> | | AG News| | BoolQ| | IMDB| | Avg. | |
> |----------|-----------|------------|------------|------------|------------|------------|------------|------------|
> | | zero-shot| few-shot | zero-shot| few-shot |zero-shot| few-shot |zero-shot| few-shot |
> |All prompts|79.0|85.6|71.0|76.5|91.2|91.4|80.4|84.5|
> |only ChatGPT prompts|74.3|85.9|67.0|76.4|90.7|90.9|77.3|84.4|
>
> >Q10. Please add the Broader Impact Statement in the final draft.
>
> I have updated Broader Impact Statement in the new version of our paper.

---

### Review · Reviewer_xCNt · 2024-09-20

**Summary Of Contributions:**

The paper proposes a process to train a small language model to design prompts. Prompts are intermediate instructions given to the “target LLM” in order to generate a better answer to the original user query. The main idea is to use an offline RL algorithm to progressively tune the prompt model. Namely, they collect data via a multi-loop process where they augment the dataset in every loop by querying the target LLM with the newly created prompts. The new data is then used to fine-tune the prompt model (offline RL), and so on.
I view the main contribution as the design of this data augmentation process and its evaluation on several language understanding tasks and against many strong baselines (to the best of my knowledge).

**Audience:**

Yes

**Broader Impact Concerns:**

Nothing to declare. This is a paper on a method to get better performance from LLMs without fine-tuning of the parameters. LLMs have a broad impact but this paper does not raise a flag.

**Claims And Evidence:**

No

**Requested Changes:**

Add the details on the "offline RL" algorithm, for clarity but also for reproducibility of the results. For now, I tick 'No' in 'Claims and Evidence' below as I don't think the paper is reproducible as is.

**Strengths And Weaknesses:**

Strengths:
* The paper is relatively easy to follow (up to some missing bits, see below)
* The empirical evaluation is thorough and shows good results

Main questions / concerns:

* The actual offline RL approach is not described. The reward loss is described, but then, how it’s used to fine tune the model seems to be implicit. But Offline RL can mean many things and the specific setting used here (single-step decision making process) would probably affect the learning algorithm used. Your task looks like an offline contextual bandit problem, is it what it actually is?

* For the reward definition, you write that you use the “Min-Max Normalization Method” but there is no citation nor explanation. The equation below (Eq 4) does  not look like a normalization to me, it’s just a sum of 2 terms. I am not sure what you meant.

Minor comments:

* It is fairly clear that an LLM was used to polish the paper, which is totally fine indeed. Just a couple of remarks: “In the realm of prompt engineering” sounds quite funny, also you used about 15 times the verb “utilize”, which is most of the time a synonym of “use” and could be replaced.
* The acronym SFT seem to never be introduced.
* Page 5: “previous researches”  is incorrect -> previous research
* The “accuracy” that you report in all the tables in the Experiments section is never really defined. I think you mean it’s the accuracy mentioned in the reward design section on page 4, but I have a doubt. It might be worth specifying exactly what you report.

---

> ### Author Response · Authors · 2024-11-07
> **Rebuttal by Authors**
>
> We sincerely thank the reviewer for the thoughtful feedback that will surely turn our paper into a better shape. We offer our responses to address your concerns as follows.
>
> >Q1.The actual offline RL approach is not described. Your task looks like an offline contextual bandit problem, is it what it actually is?
>
> We adopt the Decision Transformer (DT) training paradigm to fine-tune the PLM, as this method itself is based on the autoregressive prediction approach of language models, aligning with the prediction style of prompt generation. Specifically, we input the reward as the return-to-go (RTG), and the complete query as the state, and allow the model to autoregressively predict the entire prompt's tokens as an action, training it through $L_{prompt}$ loss function. The reward prediction loss does not directly contribute to reinforcement learning, as its optimization objective is not to generate actions. Instead, it functions similarly to an autoencoder, learning a latent representation of the reward by embedding and reconstructing the original input reward signal.
>
> Our single-step decision making process indeed resembles the offline contextual bandit setup, as our actions do not impact the environment and are generated based on the query (context).
>
> We have added these clarifications to the paper at paragraph "Reinforcement Learning Formulation", "Training Objective" and Appendix A.2. Tanks for your insightful question.
>
>
>
> >Q2. “Min-Max Normalization Method”  is no citation nor explanation.
>
> Thank you for highlighting this issue. The formula for Min-Max normalization is as follows,
> $$ Normalization(R)=\frac{R-min(R)}{max(R)-min(R)}*(100-0)$$
> it is applied to the reward values after reward summation in Equation 4. This maps the various reward ranges of different tasks to a 0-100 scale for experimental convenience. Since this is not a core aspect of our method, we did not include the formula in the paper to save space. In the revised version, we have added a citation for Min-Max normalization to help readers understand this technique in detail.
>
> >Q3. Minor comments
>
> Thank you for pointing out these issues. We have made revisions in the updated version of the paper. Specifically, we changed "In the realm of prompt engineering" to "In the domain of prompt engineering"; in the Experiments and Appendix, we added the full term "supervised fine-tuning" when SFT is mentioned for the first time; and we revised “previous research”. In the experimental section, accuracy refers to the ratio of correctly answered queries by the LLM (after prompting) to the total number of asked queries. We also added the definition of accuracy in the Experimental Setup Section of the paper.

---

### Review · Reviewer_rn9Z · 2024-11-03

**Summary Of Contributions:**

This paper introduces QPO (Query-dependent Prompt Optimization), a novel query-dependent prompt optimization framework for LLMs. QPO leverages multi-loop offline reinforcement learning and a novel Multi-Loop Augmentation technique to iteratively fine-tune a prompt generation policy. The authors show improved performance across diverse NLP and math tasks in both zero-shot and few-shot settings, showcasing QPO's efficacy, cost-efficiency, and cross-model generalizability for improved prompt engineering.

**Audience:**

Yes

**Claims And Evidence:**

No

**Requested Changes:**

1. Please upload the dataset and codebase.
2. All citations mentioned in the weaknesses section.
3. If possible, please try to use the most updated model to collect the dataset. If not, explain why the paper did not use ChatGPT4 at the first place (even gpt4 is referred as “legacy model” by openai)

**Strengths And Weaknesses:**

Strength:
1. This paper is well written and easy to follow.
2. This paper is presented in a clean and elegant way.

Weaknesses:

1. Missing details on the offline dataset. The authors did not clearly explain how the offline dataset is constructed, making the quality of the experiments less trustworthy
2. No code / dataset uploaded. With all missing details in dataset, the author did not upload the code & offline dataset, which makes the results even less trustworthy.
3. Incorrect citations. This paper have some basic misuses of citations, which the reviewer considers as a red flag for insufficient literature review and not respecting prior efforts. For example,
- When mentioning Chain-of-Thought, in page 7, paragraph baseline, the authors cited paper by Kojima et al, but not the most well known CoT paper by Wei et al
- When mentioning GPT-3.5-turbo, the author cited Ouyang et al, 2022, which is instructGPT, not GPT-3.5 turbo
- This paper applies offline RL in training, but no related papers about the offline RL field is mentioned, the author should at least mention [2], when talking about offline RL.
4. The models being used in the paper are far from the most updated versions. For example, in page 7 implementation details, why not use ChatGPT-4 (or even 4o)?



[1] Wei, Jason, et al. "Chain-of-thought prompting elicits reasoning in large language models." Advances in neural information processing systems 35 (2022): 24824-24837.

[2] Levine, Sergey, et al. "Offline reinforcement learning: Tutorial, review, and perspectives on open problems." arXiv preprint arXiv:2005.01643 (2020).

---

> ### Author Response · Authors · 2024-11-07
> **Rebuttal by Authors**
>
> We sincerely thank the reviewer for the thoughtful feedback that will surely turn our paper into a better shape. We offer our responses to address your concerns as follows.
>
> >Q1. how the offline dataset is constructed
>
> The specific steps for constructing the offline dataset are as follows, which is also visualized in Figure 2 at "Step0: Dataset Construction":
>
> First, sample queries from the task dataset, then combine them with the collected prompts and input the query-prompt pair into the target LLM to obtain answers. Then, using our designed reward function, we calculate the reward value for each query-prompt pair. These query-prompt-reward triplets thus form the samples in our offline dataset.
>
> Thank you for raising this question; we have added the clearer clarification in the revised version of the paper in paragraph Dataset Construction in Section 2.3.1.
>
> >Q2. No code / dataset uploaded
>
> We provide all datasets and code at the anonymous link：https://anonymous.4open.science/r/QPO-7453/README.md
>
> >Q3. Incorrect citations.
>
> Thank you for highlighting these issues. We have revised these citation issues in the paper.
>
> >Q4. Use the most updated model to collect the dataset.
>
> Our initial choice of GPT-3.5 was to demonstrate that our method is applicable to high-performance APIs while minimizing budget requirements. To address your concern, we have now conducted experiments using $\textbf{GPT-4o}$, but due to budget constraints, we limited the experiments to the GSM8K dataset only. The results are shown in the table below. As demonstrated, our method also outperforms other baselines on most updated models. This experiment has been included in Appendix C.2 of the paper. Thank you for your question.
>
> | | GSM8K|
> |----------|-----------|
> |CoT|95.8|
> |ChatGPT|94.7|
> |APE|95.0|
> |Prompt-OIRL| 93.5|
> |QPO|96.5|

---

### Author Response · Authors · 2024-11-28

Dear Action Editor and Reviewers,

We are sincerely grateful to the reviewers for dedicating their time and offering valuable feedback. In our rebuttal, we have replied to each reviewer's questions in detail, and we would like to summarize our rebuttal and revisions to our manuscript in this official comment.

* Following the comment of Reviewer **xCNt**, we expanded our discussion on the Offline RL algorithm, specifically detailing the integration of the Decision Transformer (DT) training methodology for fine-tuning the pretrained language model, which is supplemented in Section 2.2 of the manuscript.

* Following the comment of Reviewer **rn9Z**, we supplemented the detailed process of constructing the offline dataset in Section 2.3.1 and changed the paragraph titile from "Data Availability" to "Dataset Construction".

* Following the comment of Reviewer **giNB**, we supplemented the explanation of autoencoder in Appendix A.4 in our paper.

* Other revisions: We corrected the citations pointed out by Reviewer **rn9Z**, added a citation for the Min-Max Normalization method suggested by Reviewer **xCNt**, made revisions based on the Minor Comments proposed by Reviewer **xCNt**, and supplemented the experimental implementation details that were missing in the paper as requested by Reviewer **giNB**. The specific locations of these modifications in the paper are all mentioned in the individual responses to each reviewer.

* We provided our offline dataset and the source code in the supplementary material by the request of Reviewer **rn9Z**.

Concerning our new experiments:

* By the request of Reviewer **giNB**, we added three new ablation studies on our reward design and training objective, including the contributions of query-level and task-level rewards, the importance of perplexity penalty in query-level reward, and the comparison with different implementation of reward prediction loss. All the results and analyses can be found in detail in Appendix C.4 of the new manuscript.

* In addition, by the request of Reviewer **giNB**, we added ablation study on the prompt quality in our dataset. We only use the rewritten prompts without initial high-quality prompts to train the policy model, and results show that QPO is robust to prompt quality. The detailed results and analyses are shown in Appendix C.6 in our paper.

* By the request of Reviewer **rn9Z**, we added new results on GPT-4o, the most updated model, to show that QPO is applicable to models with various capabilities. These results are shown in Appendix C.2 in our paper.

Overall, we hope that our answers helped clarify the unclear points, as well as strengthen our arguments and empirical validation.

---

### Decision · Action_Editor_zc4y · 2025-02-04

**Recommendation:** Accept as is

**Comment:**

The paper touches on an interesting problem to the community, that is, how to do efficient prompt engineering. The paper provides details and experiments to support the claims. The method is easy to follow and reproducible. The authors did a great job in improving the paper during the rebuttal process.

**Audience:**

The paper is interested in a wide audience interested in LLMs and prompt engineering for LLMs.

**Claims And Evidence:**

The paper proposes QPO :  Query-dependent Prompt Optimization via
Multi-Loop Offline Reinforcement Learning. It leverages multi-loop offline
reinforcement learning to iteratively fine-tune a small pretrained language model to generate
optimal prompts tailored to the input queries, thus significantly improving the prompting effect on the large target LLM.  The authors claim that these iterative loops bootstrap the model towards generating optimal
prompts. Experiments on various LLM scales and diverse NLP and math tasks demonstrate
the efficacy and cost-efficiency of the method in both zero-shot and few-shot scenarios. The claims are supported by some experiments subject to limits of compute on the authors side. The paper also includes details of dataset generation and is accompanied by the code.